# Cobalt Serum Level as a Biomarker of Cause-Specific Survival among Prostate Cancer Patients

**DOI:** 10.3390/cancers16152618

**Published:** 2024-07-23

**Authors:** Sandra Pietrzak, Wojciech Marciniak, Róża Derkacz, Milena Matuszczak, Adam Kiljańczyk, Piotr Baszuk, Marta Bryśkiewicz, Andrzej Sikorski, Jacek Gronwald, Marcin Słojewski, Cezary Cybulski, Adam Gołąb, Tomasz Huzarski, Tadeusz Dębniak, Marcin R. Lener, Anna Jakubowska, Tomasz Kluz, Marianna Soroka, Rodney J. Scott, Jan Lubiński

**Affiliations:** 1Department of Genetics and Pathology, International Hereditary Cancer Center, Pomeranian Medical University in Szczecin, ul. Unii Lubelskiej 1, 71-252 Szczecin, Poland; sandra.pietrzak@pum.edu.pl (S.P.); milena.matuszczak@pum.edu.pl (M.M.); adam.kiljanczyk@pum.edu.pl (A.K.); piotr.baszuk@pum.edu.pl (P.B.); marta.bryskiewicz@pum.edu.pl (M.B.); jacek.gronwald@pum.edu.pl (J.G.); cezarycy@pum.edu.pl (C.C.); tomasz.huzarski@pum.edu.pl (T.H.); tadeusz.debniak@pum.edu.pl (T.D.); marcin.lener@pum.edu.pl (M.R.L.); anna.jakubowska@pum.edu.pl (A.J.); 2Read-Gene, Grzepnica, ul. Alabastrowa 8, 72-003 Dobra (Szczecińska), Poland; wojciech.marciniak@read-gene.com (W.M.); roza.derkacz@read-gene.com (R.D.); 3Department of Urology and Urological Oncology, Pomeranian Medical University in Szczecin, al. Powstańców Wielkopolskich 72, 71-899 Szczecin, Poland; sikor@post.pl (A.S.); marcin.slojewski@pum.edu.pl (M.S.); adam.golab@pum.edu.pl (A.G.); 4Department of Clinical Genetics and Pathology, University of Zielona Góra, ul. Zyty 28, 65-046 Zielona Góra, Poland; 5Department of Gynecology, Gynecology Oncology and Obstetrics, Fryderyk Chopin University Hospital No. 1, ul. Szopena 2, 35-055 Rzeszow, Poland; jtkluz@interia.pl; 6Institute of Medical Sciences, Medical College of Rzeszow University, al. Rejtana 16c, 35-959 Rzeszow, Poland; 7Department of Genetics and Genomics, Institute of Biology, University of Szczecin, ul. Felczaka 3c, 71-412 Szczecin, Poland; marianna.soroka@usz.edu.pl; 8Priority Research Centre for Cancer Research, Innovation and Translation, Hunter Medical Research Institute, New Lambton, NSW 2305, Australia; rodney.scott@newcastle.edu.au; 9School of Biomedical Sciences and Pharmacy, Faculty of Health and Medicine, University of Newcastle, Callaghan, NSW 2308, Australia; 10Division of Molecular Medicine, Pathology North, John Hunter Hospital, New Lambton, NSW 2305, Australia

**Keywords:** prostate cancer, cobalt, survival, biomarker

## Abstract

**Simple Summary:**

Prostate cancer is the most common cancer detected among men and it is the second leading cause of death. According to the WHO, cobalt is probably involved in carcinogenesis. However, there are no studies related to cobalt levels and survival in prostate cancer patients. The aim of this prospective study was to investigate the relationship between serum cobalt levels and survival among prostate cancer patients taking into consideration prostate cancer-specific deaths and non-cancer causes of death. Our findings, based on 261 Polish prostate cancer patients, show that individuals with high serum cobalt levels have a significantly worse survival compared to participants with low serum cobalt levels.

**Abstract:**

Prostate cancer is the most common cancer diagnosed in men and the second leading cause of death in male cancer patients. The WHO suggests that cobalt is involved in the carcinogenesis of prostate cancer. There are, however, no studies associating cobalt levels and prostate cancer patient survival. In this study, 261 Polish prostate cancer (*n* = 261) patients were recruited into a prospective cohort between 2009 and 2015. Serum cobalt levels were measured using ICP-MS after prostate cancer diagnosis and before treatment. All study participants were assigned into quartiles (QI-QIV) based on the distribution of serum cobalt levels among censored patients. Univariable and multivariable COX regression models were used to calculate hazard ratios (HRs) for each serum cobalt level quartile. We found a significant relationship between high serum cobalt levels and poor prostate cancer patient total survival (HR = 2.60; 95% CI: 1.17–5.82; *p* = 0.02). In relation to prostate cancer patients who died as a result of other non-cancer causes, the association with high levels of cobalt was even stronger (HR = 3.67; 95% CI: 1.03–13.00; *p* = 0.04). The impact of high serum cobalt levels on overall survival of prostate cancer-specific-related deaths was not statistically significant.

## 1. Introduction

Prostate cancer is the prevailing malignancy among men and ranks as the second most frequent cause of cancer-related death. Globally, there were more than 1.4 million new cases and 375,304 deaths from prostate cancer in 2020 [1]. It is estimated in the USA, there will be, in 2024, 299,010 new prostate cancer diagnoses and 35,250 deaths. Prostate cancer represents 14.90% of all new cancer cases in the U.S. [2]. The estimated lifetime risk of developing prostate cancer is 11%, and the risk of death is approximately 2% [3]. The NIH “Cancer Stat Facts” for Prostate Cancer reports that 5-year relative survival between 2014 and 2020 was 97.50% [2].

A significant problem in the management of prostate cancer is the identification of determinants that affect survival. The prognosis affecting the survival of patients with prostate cancer includes various clinical, pathological, and biological factors. Crucial predictors include the stage of disease, the levels of prostate-specific antigen (PSA), the degree of histological differentiation of the tumor (assessed by the Gleason score), the presence of metastases, and the patient’s general condition (assessed by the ECOG or the Karnofsky scale). High PSA values and a high Gleason score are associated with a poorer prognosis. Clinical assessment according to the TNM (tumor, node, metastasis) classification is also key—patients with lymph node metastasis or distant metastasis tend to have a worse prognosis. In addition, genetic and molecular studies have shown that the presence of mutations, such as those in *PALB2*, *BRCA2*, *ATM*, and *NBN* [4,5], or changes in certain signaling pathways, can affect tumor aggressiveness and response to treatment, which in turn affects patient survival.

The main causes of death among cancer patients are widespread metastases, cancer-associated thrombosis (CAT) [6], multi-organ failure, infarction (acute hypoxemia), respiratory failure, infection (i.e., pneumonia, septicemia, and peritonitis), carcinomatosis, and hemorrhage. Some of the factors affecting survival include disease progression, staging, grading, age, hypercalcemia, respiratory insufficiency, obesity, anemia, pulmonary or renal disease, co-morbid illness, cancer treatment side effects (e.g., chemotherapy-induced cardiomyopathy), surgery with general anesthesia, immunosuppression, multidrug resistance (MDR), prolonged immobilization, inherited thrombophilia factors, and hospitalization [7]. Among environmental factors influencing survival, some chemical elements have been suggested. Recently, we were able to find an association between prostate cancer survival and blood levels of selenium, zinc, and copper [6].

The influence of cobalt on survival involves oxidative DNA damage, interference with DNA repair, and direct genotoxic interactions with DNA, increasing cancer risk aggressiveness. Cobalt interferes with DNA repair processes, inhibiting the repair of DNA double-strand breaks (DSBs) and hindering the activity of key repair proteins. This leads to persistent DNA damage and genomic instability. Cobalt also generates oxidative stress by catalyzing the production of reactive oxygen species (ROS). Cobalt can bind directly to DNA, causing structural alterations that interfere with DNA replication and transcription, inducing genotoxic effects, like chromosomal aberrations and micronuclei formation.

The effect of cobalt on survival among patients with prostate cancer has not been studied, and little is known about how it may influence disease outcomes. The aim of this study is to determine whether cobalt is a key factor driving poor outcomes for prostate cancer patients.

## 2. Materials and Methods

### 2.1. Study Group

A total of 261 consecutively enrolled men with prostate cancer formed the study population. All patients were recruited after disease diagnosis based on histopathological examination at the Department of Urology and the Urological Clinical Hospital of the Pomeranian Medical University in Szczecin, between 2009 and 2015. This study was conducted in accordance with the Helsinki Declaration and the Ethics Committee of Pomeranian Medical University in Szczecin (KB-0012/73/10, 21 June 2010). All participants provided written informed consent prior to enrollment in this study.

### 2.2. Measurement Methodology

Blood for serum was taken in a fasting state by venipuncture using a Vacutainer^®^ System. After collection, the tubes were incubated at room temperature for a minimum of 30 min to clot, and after this time, the tubes were centrifuged in 1300 G for 12 min. After centrifugation, serum was aliquoted and transferred into new cryovials and deep freezed (−80 °C) until analysis.

Serum cobalt (^59^Co) levels were quantified by inductively coupled mass spectrometry (ICP-MS, NexION 350D) using KED mode (Kinetic Energy Discrimination), achieving a reduction in polyatomic interferences. The spectrometer was calibrated using the matrix-matched calibration technique. Calibration standards were prepared fresh daily from 10 µg/mL Multi-Element Calibration Standard 3 by diluting with a blank reagent to final concentrations of 0.10, 0.20, 0.30, 0.50, and 1.00 µg/L. All calibration curves have correlation coefficients greater than 0.999. Internal standardization, using a Rhodium isotope, was used to eliminate matrix-related effects and instrument drift.

Samples were diluted 30 times in an alkali buffer. A dilution buffer consisted of high-purity water (>18 MΩ), TMAH, Triton X-100, ethanol, and EDTA.

Accuracy and precision of measurements were assessed using certified reference material (CRM), Clincheck Plasmonorm Serum Trace Elements Level 1, and Seronorm Serum Trace Elements. Data from the manufacturers of the reagents and devices used to determine cobalt levels are shown in Appendix A (Table A1). Technical details, plasma operating settings, and mass spectrometer acquisition parameters are available upon request.

### 2.3. Statistical Analysis

#### 2.3.1. Descriptive Analysis

All prostate cancer patients (*n* = 261) were assigned to one of four quartiles (QI-QIV), determined by the serum cobalt distribution among alive subjects (*n* = 205). The selected reference category was the group with the lowest serum cobalt levels (QI) associated with the lowest death/alive ratio (8/51). All qualitative variables were described by nominal values and percentages. Quantitative variables were described by the range and mean concentrations of cobalt. The relationship between serum cobalt levels and clinical factors was compared using the nonparametric Kruskal–Wallis test for cobalt levels expressed quantitatively, respectively.

#### 2.3.2. Univariable Analysis

In order to estimate the relationship between serum cobalt levels on prostate cancer survival, COX proportional hazard regression models were calculated. This method is the most frequently used approach used to investigate the association between time to event outcome and a set of explanatory variables.

In order to present the survival rate in time, depending on serum cobalt levels divided into categories (QI–QIV), Kaplan–Meier curves were generated, and log-rank tests were performed.

#### 2.3.3. Multivariable Analysis

Multivariable COX proportional hazard models take into account the following variables: age of diagnosis (≤60/>60), Gleason (<7/7/>7), PSA (<4/4–10/>10), and vital status during follow up (alive/dead). A follow-up time of >5 years was recognized as the exact 5-year observation time.

#### 2.3.4. Software

All calculations and graphics were performed in an R statistical environment (R: A language and environment for statistical computing; R Foundation for Statistical Computing, Vienna, Austria 2023; R version: 4.3.2).

## 3. Results

The characteristics of the study group and the relationship between the serum cobalt level and clinical factors, regardless of the cause of death, are shown in Table 1.

The mean, interquartile range (IQR), and median cobalt levels for the following variables status (alive/dead), age (≤60, >60), Gleason (<7, 7, >7), PSA (<4, 4–10, >10), prostatectomy (no/yes), radiotherapy (no/yes), chemotherapy (no/yes), and hormonotherapy (no/yes) are presented in Table 2. Differences among the medians were compared using Kruskal–Wallis tests as well. Described analyses were performed among all study participants (*n* = 261), prostate cancer patients with non-cancer-related death (*n* = 230), and prostate cancer-specific causes of death (*n* = 236) separately. 

The mean serum cobalt level for the whole study group was 0.16 µg/L, and the median serum cobalt level was 0.13 µg/L. The median serum cobalt level was statistically different for the following variables: vital status (alive/dead)—*p* < 0.01; Gleason (<7, 7, >7)—*p* < 0.01, prostatectomy (no/yes)—*p* = 0.03, and hormonotherapy (no/yes)—*p* = 0.03. The median serum cobalt levels were higher for deceased patients (0.16 µg/L) compared to the alive study participants (0.12 µg/L). The median serum cobalt levels for Gleason (<7, 7, >7) were 0.15 µg/L, 0.12 µg/L, and 0.13 µg/L, respectively. Prostate cancer patients after prostatectomy have lower median serum cobalt levels (0.12 µg/L) compared to patients in which prostatectomy was not performed (0.15 µg/L). The median serum cobalt levels were higher among patients who received the hormonotherapy (0.13 µg/L) compared to participants to which hormonotherapy was not applied (0.12 µg/L). Similar observations in relation to the difference in serum cobalt levels among aforementioned clinical factors were observed for a subgroup of prostate cancer patients with non-cancer-related death (*n* = 230) and prostate cancer-specific causes of death (*n* = 236) as well.

For the entire study group (*n* = 261), prostate cancer patients with high cobalt levels (QIV) had worse survival compared to those with the lowest cobalt serum levels (QI) in both univariable (HR = 3.02; 95% CI: 1.38–6.59; *p* < 0.01) and multivariable (HR = 2.60; 95% CI: 1.17–5.82; *p* = 0.02) COX regression models. The results for the univariable and multivariable approaches are shown in Table 3.

Similar analyses were performed for the subgroup of prostate cancer patients with non-cancer-related deaths (*n* = 230).

In this subgroup, the association between high cobalt serum levels and worse survival was even stronger. The hazard ratio of death among prostate cancer patients with non-cancer-related death was almost four times greater for those with high cobalt serum levels: HR = 4.39; 95% CI: 1.27–15.20; *p* = 0.02 and HR = 3.67; 95% CI: 1.03–13.00; *p* = 0.04 for uni- and multivariable COX regression models, respectively (Table 4).

Similar analyses were performed for the subgroup of patients with prostate cancer-specific causes of death (*n* = 236). 

In the following subgroup, no statistically significant differences were observed in survival, depending on the cobalt serum levels (Table 5).

The survival of prostate cancer patients according to the cobalt quartile is presented in the Kaplan–Meier curves for all-cause deaths, non-cancer-related deaths, and prostate cancer-specific deaths, respectively (Figure 1, Figure 2 and Figure 3).

## 4. Discussion

The diagnosis of prostate cancer involves measuring the concentration of prostate-specific antigen (PSA) and performing a biopsy of the gland. Although the introduction of PSA testing has increased detection rates, it has not significantly reduced mortality due to overdiagnosis and overtreatment. Currently, most prostate cancers are detected at an asymptomatic stage, with patients diagnosed due to elevated PSA levels or abnormal digital rectal examination. Some patients present with lower urinary tract symptoms (LUTSs) when cancer develops in the central part of the prostate. Elevated PSA levels suggest the presence of cancer, but approximately 25% of men with elevated PSA do not have cancer, and nearly 20% of prostate cancer patients have normal PSA levels [8].

There are newer, more effective biomarkers that can more accurately assess the presence and prognosis of cancer [9,10]. Some of these assessments have high precision for clinically significant prostate cancer (csPCa) [11]. The validity of PSA screening is questioned due to the unnecessary detection of clinically insignificant cancers, which would never cause complications or death. It has been shown that to prevent one prostate cancer death, over a thousand men need to be screened, and an additional 37 cancers need to be detected, which does not impact overall mortality risk [8].

About 5–9% of prostate cancers are detected at an advanced stage [8]. These cancers primarily metastase to bones, causing symptoms that include pain, pressure on sensitive structures, or fractures. The primary goal of prostate cancer diagnosis is to accurately assess the risk of disease progression and cancer-specific mortality [12]. Improving diagnostic methods is essential to distinguish benign cases from those with a high risk of progression, optimize care, and personalize treatment. Treatment strategies include surgery, radiotherapy (external beam radiation and/or brachytherapy), hormone therapy, and active surveillance, depending on the stage and risk of progression.

Moreover, there are new therapies for prostate cancer offering a number of innovative approaches that can significantly improve treatment efficacy and reduce side effects. Among them, we can distinguish focal cryotherapy [13], transurethral ultrasound ablation [14,15], transperineal laser ablation [15], photothermal therapy [16], photodynamic therapy [17], nanotherapeutics [18], autoantibodies (AAbs) [19], tumor-associated macrophages (TAMs) [20], and CAR T cells. In addition, shaping the gut microbiome to enhance the response to checkpoint inhibitors (ICIs) through probiotic therapy and fecal microbiota transplantation has the potential to enhance the immune response [21]. The synergy of autophagy inhibition and a PD-L1 blockade via an acid-sensitive nanoparticle (P-PDL1-CP) may also enhance the efficacy of prostate cancer immunotherapy [22]. 

The most commonly performed imaging test for the prostate gland is transrectal ultrasonography (TRUS). It is a non-invasive, inexpensive, and readily available method, but its value in detecting cancer and assessing the stage is limited and dependent on the examiner’s experience. Imaging methods, like magnetic resonance imaging (MRI) and bone scintigraphy, help assess disease progression. PI-RADS (Prostate Imaging–Reporting and Data System) is a structured reporting scheme for multiparametric prostate MRI (mpMRI) used in the evaluation of suspected prostate cancer in treatment-naive prostate glands. This system combines imaging findings from T2-weighted imaging (T2WI), diffusion-weighted imaging (DWI), and dynamic contrast enhancement (DCE) to predict the probability of clinically significant cancer. Clinically significant cancer is defined by factors such as a Gleason score ≥ 7, tumor volume > 0.5 mL, or extraprostatic extension. Each lesion is assigned a PI-RADS score from 1 to 5, indicating the likelihood of clinically significant cancer, with scores of 4 or 5 suggesting a higher probability and typically prompting consideration of a biopsy.

Current efforts focus on developing even better and more effective methods for accurately assessing disease stages. One such method is prostate cancer mapping using multiparametric MRI (mpMRI) [23]. Comparing the diagnostic effectiveness of mpMRI and biparametric MRI (bpMRI), studies have shown that bpMRI is comparable to mpMRI in detecting prostate cancer (PCa) and clinically significant prostate cancer (csPCa) [24]. 

PSMA (prostate-specific membrane antigen)–PET–CT [25] is becoming increasingly popular in diagnosing and assessing the stage of prostate cancer due to its high accuracy. PSMA is associated with aggressive prostate cancer and advanced disease, and its high expression correlates with worse survival outcomes. PSMA ligand uptake in the primary tumor correlates with traditional prognostic factors, such as high Gleason scores and reduced progression-free survival after radical prostatectomy. In recent years, the clinical utility of PSMA contrast agents in molecular imaging has been particularly notable. Among these, we can highlight [64Cu]Cu-PSMA-617 [26], [99mTc]Tc-PSMA-I&S [27], [177Lu]Lu-PSMA-617 [28], [18F]DCFPyL [29], and [68Ga]Ga-PSMA-11 PET/MRI [30].

Risk factors for prostate cancer include several key elements affecting patient prognosis and survival. The diagnosis is based on the histopathological examination of the biopsy, with the most important prognostic factors being the Gleason score, PSA levels (especially high levels and rapid doubling time), and the degree of local advancement. The Gleason score assesses the aggressiveness of the tumor based on the microscopic appearance of the cells. High PSA levels post-surgery are being studied as indicators of disease progression and cancer-specific survival. 

Recent reports have shown a growing interest in persistent PSA levels after surgery as a possible additional indicator of disease progression and cancer-specific survival. A study [31] evaluated the association between PSA persistence and long-term oncological outcomes in prostate cancer risk groups. They found that persistent levels of this biomarker can be used as an independent indicator of worse long-term outcomes in high-risk prostate cancer patients. In contrast, among intermediate-risk patients, this parameter significantly predicts only biochemical recurrence, with no effect on outcomes in low-risk prostate cancer patients. Due to the problem of overtreatment, researchers are striving to discover more precise prognostic methods.

Oligometastatic prostate cancer (omPCa) [32,33] is an intermediate state with a limited number of metastases and specific locations. Accurate staging is crucial for detecting oligometastases, which is facilitated by PSMA-PET. Cytoreductive radical prostatectomy can provide survival benefits for selected patients with omPCa.

Chemical elements that affect survival include essential elements, such as selenium, zinc, and iron, and carcinogenic elements, which include cadmium and lead. The effects of various elements on progression have already been described in our previous report [34]. One of the probable carcinogens is the essential micronutrient—cobalt.

According to the WHO, cobalt is probably involved in carcinogenesis [35]. To the best of our knowledge, there are no studies associating the survival of cancer patients and cobalt levels. Herein, we addressed the question as to whether cobalt can influence survival. Cobalt in its organic form is embedded in the ring of hydroxycobalamin (vitamin B12), which plays a biologically essential role. Cobalt is mainly distributed in the serum, liver, kidney, heart, and spleen. Smaller amounts of cobalt can be also detected in the skeleton, hair, lymphatic vessels, and pancreas [36]. Blood levels of cobalt range from 0.10 to 1.20 µg/L (mean 0.50 µg/L) in a group with no occupational exposure [37]. Cobalt levels in red blood cells (RBCs) remain longer than in serum due to cobalt’s permanent binding to blood cells, but its levels are higher in serum as a result of its higher affinity for plasma albumin [38]. In addition, cobalt can also be measured in urine, but its levels are subject to the most dynamic changes; for this reason, serum testing is recommended [38]. Exposure to excessive amounts has various negative health effects that are attributed to occupational, environmental, dietary, and medical sources. There appear to be two categories of exposure: high (occupational, excessive supplementation) and low (cobalt exposure in the general population, e.g., our patients) [39]. Until now, studies have focused on very strong exposures leading to high blood levels of >300 µg. Such levels have toxicities that are well documented in the literature, including neurological (e.g., hearing and vision impairment), endocrine, and cardiovascular deficits [36]. Cobalt interferes with DNA repair and causes DNA damage. Cobalt toxicity primarily arises from its ability to generate reactive oxygen species (ROS) within cells, leading to significant intracellular damage. When cobalt ions (Co^2+^) interact with hydrogen peroxide (H_2_O_2_) and ascorbate, they produce ROS, such as hydroxyl radicals (•OH), superoxide anion (O_2_•^−^), and hydroperoxyl radicals (•OOH). These ROS induce oxidative stress, causing DNA damage, impairing cellular functions, and triggering mitochondrial dysfunction. Additionally, cobalt exposure results in the upregulation of antioxidant genes, but excessive ROS can overwhelm these defenses, leading to apoptosis and cell cycle arrest [40,41]. Serum ferritin levels are strongly inversely correlated with blood cobalt levels. In addition, there is an inverse correlation between iron and cobalt, which is more pronounced in women [42]. Higher cobalt levels are observed in people with reduced ferritin as well as total iron levels, and this may be due to a common absorption pathway in the intestine where cobalt can displace iron and compete for uptake by the metal tracer DMT-1 and the NRAMP1 protein [42]. Excess cobalt that can induce increased ROS production may be linked to mitochondrial alteration and cellular death, such as ferroptosis [43].

Cobalt is removed via the kidneys, and older patients with renal impairment have an increased susceptibility to increased cobalt compared to young and healthy people exposed to the same factors [38]. Because of their greater predisposition to accumulate this element, older patients may have an increased risk of death from organ failure due to higher levels of cobalt. In addition, it seems likely that since high doses of cobalt increase hematocrit, hemoglobin, and RBCs and may lead to polycythemia, which predisposes to thrombosis, it is one of the most common causes of non-cancer death [38,44]. Depending on the source of cobalt, there is an additional risk of lung diseases, such as diffuse lung fibrosis and bronchial asthma, which increase the risk of respiratory failure, contributing to increased mortality [45].

Foods contain cobalt in various concentrations [46]. Clams, seaweeds, animal livers, dried enriched yeast, fish, nuts, coffee, meat, chocolate, butter, green leafy vegetables, and cereals contain the highest average concentration of cobalt; moreover, food processing also contributes to increased cobalt concentration [47,48,49,50,51]. The FDA has not made clear recommendations on the daily intake of cobalt; The European Food Safety Authority has suggested a safe dose of 600 µg/day [52]. Manufacturers recommend the intake of supplements containing 1 mg Co/day to improve red cell production, protein synthesis, fat and carbohydrate metabolism, and myelin sheath repair in the central nervous system [53], and this also makes them effective blood doping agents used by athletes [54,55]. Vitamin B12 is found mainly in meat and dairy products [36]. The recommended daily intake of vitamin B12 for adults is 3 µg/day, which provides 0.01 µg of cobalt [56]. Foods and energy drinks can be fortified with large amounts of vitamin B12 [57].

The consumed dose, solubility of the compound, overnight fasting, and nutritional status (increased absorption of cobalt in subjects with iron deficiency [25]) contribute to extreme inter-individual variability (5–97%) of gastrointestinal cobalt absorption [36,42,58,59,60,61,62,63]. Due to the increased demand for iron and shared uptake mechanisms, women assimilate increased amounts of cobalt and, due to the reduced renal excretion of cobalt (3.50 mL/min for women vs. 5.50 mL/min for men), have an additional tendency to accumulate it [38]. In addition, its accumulation is variable within the same sex (menstruating women with longer cycles have a higher demand), according to physical activity (people with active lifestyles have more frequent erythrocyte exchange and an increased excretion of iron in urine and sweat, such that they absorb larger amounts of these elements than people with sedentary lifestyles) [38]. Ionic forms of cobalt (Co^2+^ and Co^3+^) demonstrate no variable absorption from the gastrointestinal tract [64]. After absorption (from all routes of uptake), cobalt is distributed mainly to the serum, whole blood, liver, kidney, heart, and spleen, such that urine and blood appear to be useful sources for measurement [59,65,66,67,68,69,70,71,72,73]. The total amount of cobalt in the form of vitamin B12 from food in the adult human body is about 0.25 mg, of which 50–90% is found in the liver [65].

In the past, cobalt compounds were used to treat anemia. Average doses ranged from 25 to 150 mg/day of CoCl_2_ but could be as high as 300 mg/day [65,74,75,76,77,78], which were administered for several months. The most common side effects were goiters and decreased iodine uptake [79,80], and the less frequent symptoms were optic atrophy, hearing loss, and cardiomyopathy [74,81,82,83,84,85], which were observed after discontinuation of CoCl_2_ for anemia treatment.

Several cobalt chelators have been studied, of which three appear to be the most effective: ethylendiamine–tetraacetic acid (EDTA), diethylentriamepentaacetic acid (DTPA), and N-acetyl cysteine (NAC) [44,86,87]. NAC supplementation can help reduce blood cobalt levels by about three times in less than a year, but it should be noted that the trial was conducted on a single patient [88]. In addition, in a study testing the detoxification ability of NAC, dimercaptosuccinic acid (DMSA), EDTA, and DTPA on rats, it was shown that of the chelators tested, only NAC reduces organic cobalt levels [86]. Also noteworthy is the fact that studies using NAC report no toxicity or side effects [44]. EDTA is mainly used as an adjunct therapy when a hip prosthesis gradually wears, resulting in cobalt release. Work has shown that NAC provides a short-term reduction in blood cobalt levels. However, cobalt levels rebounded within a few days [87].

We evaluated whether cobalt serum levels could be related to the survival of patients with prostate cancer. Our study cohort consisted of 261 patients, of whom 56 died during follow up, 55.36% of them died from prostate cancer, and 44.64% from other causes.

Cobalt levels > 0.10 μg/L (QI) resulted in increased all-cause mortality by as much as 2.60 folds (95% CI: 1.17–5.82; *p* = 0.02) in QIV. Furthermore, patients with higher than the aforementioned cobalt levels were at risk of increased non-cancerous-caused mortality by up to 3.67 folds (95% CI: 1.03–13.00; *p* = 0.04) in QIV. We found that survival related to the progression of prostate cancer in patients with cobalt levels of >0.10 μg/L has not changed significantly. This indicates that serum levels of this element < 0.10 μg/L may be associated with improved survival with a particularly strong relationship if only non-cancer-related deaths are considered. Therefore, it may be a valuable biomarker for the prognosis of prostate cancer. This also raises the question if the optimization of cobalt levels can improve prostate cancer survival.

Limitations in this study were the relatively small study group and relatively low numbers of events (deaths), especially in subgroups of non-cancer-related deaths (*n* = 25) and prostate cancer-specific deaths (*n* = 31). Moreover, individuals included in this study came from the same geographical region, within Poland, which may not be a direct reference to all prostate cancer patients.

## 5. Conclusions

In summary, this is the first report showing the potential significance of cobalt serum levels on the survival of prostate cancer patients. Obviously, further investigations are required to validate the influence of cobalt in prostate cancer survival and include studies of other cancer types. Our data indicate that cobalt may be one of the elements critical for improved survival (HR = 2.60; 95% CI: 1.17–5.82; *p* = 0.02 and HR = 3.67; 95% CI: 1.03–13.00; *p* = 0.04 for all causes of deaths and non-cancer-related deaths, respectively), and its reduction to appropriate physiological levels might significantly improve treatment outcomes. It may be possible in the future to also address the benefits and risks of vit. B_12_ supplementation.

## Figures and Tables

**Figure 1 cancers-16-02618-f001:**
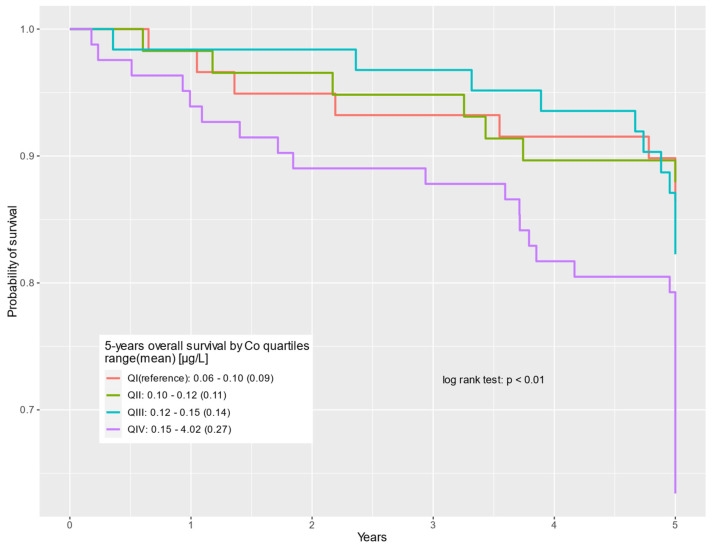
Five-year overall survival by cobalt serum levels (µg/L) for all causes of death.

**Figure 2 cancers-16-02618-f002:**
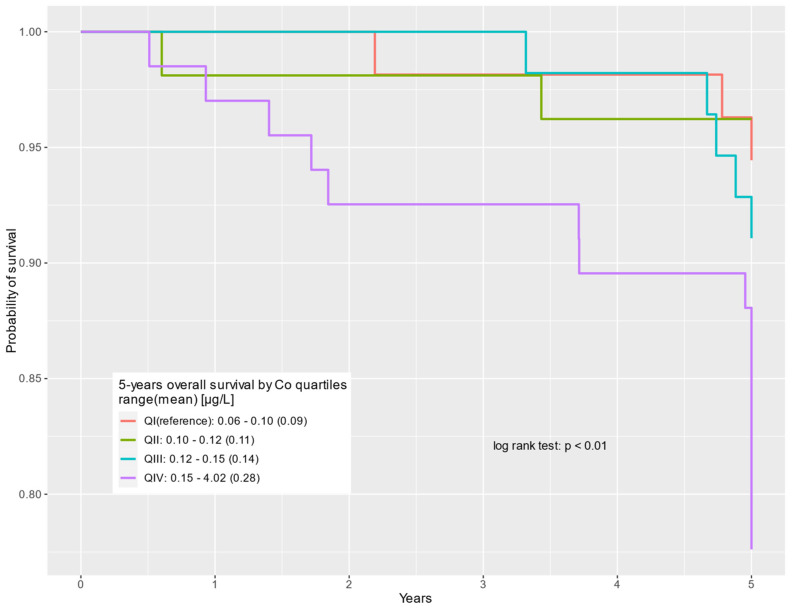
Five-year overall survival by cobalt serum levels (µg/L) for non-cancer-related deaths.

**Figure 3 cancers-16-02618-f003:**
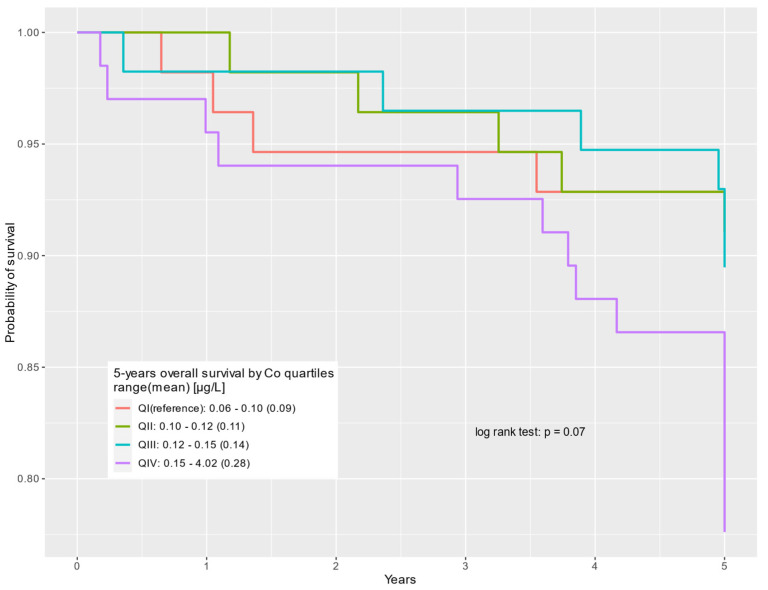
Five-year overall survival by cobalt serum levels (µg/L) for prostate cancer-specific deaths.

**Table 1 cancers-16-02618-t001:** Characteristics of the study group (*n* = 261).

Variable	Overall, *n* = 261
Vital status	
Alive	205 (78.54%)
Dead	56 (21.46%)
Cause of death	
Other	25 (44.64%)
Prostate cancer	31 (55.36%)
Age of diagnosis	46–86 (65.70)
<60	58 (22.22%)
≥60	203 (77.78%)
Gleason	
<7	85 (32.57%)
7	133 (50.96%)
>7	43 (16.48%)
PSA	0.25–700.00 (20.94)
<4	13 (4.98%)
4–10	143 (54.79%)
>10	105 (40.23%)
Prostatectomy	
No	57 (22.62%)
Yes	195 (77.38%)
Unknown	9
Radiotherapy	
No	114 (48.51%)
Yes	121 (51.49%)
Unknown	26
Chemotherapy	
No	207 (92.83%)
Yes	16 (7.17%)
Unknown	38
Hormone therapy	
No	154 (64.17%)
Yes	86 (35.83%)
Unknown	21
Cobalt level	
QI: 0.06–0.10 (0.09)	59 (22.61%)
QII: 0.10–0.12 (0.11)	58 (22.22%)
QIII: 0.12–0.15 (0.14)	62 (23.75%)
QIV: 0.15–4.02 (0.27)	82 (31.42%)

**Table 2 cancers-16-02618-t002:** Mean, IQR, and median cobalt serum levels in the associated variables.

	All Study Group (*n* = 261)					Non-Cancer-Related Deaths (*n* = 230)					Prostate Cancer-Specific Deaths (*n* = 236)				
Variable	*n* (%)	Mean	IQR	Median	*p* ^1^	*n* (%)	Mean	IQR	Median	*p* ^1^	*n* (%)	Mean	IQR	Median	*p* ^1^
Vital status					<0.01					<0.01					0.02
Alive	205 (79%)	0.16	0.06	0.12		205 (89%)	0.16	0.06	0.12		205 (87%)	0.16	0.06	0.12	
Dead	56 (21%)	0.17	0.07	0.16		25 (11%)	0.16	0.05	0.17		31 (13%)	0.17	0.09	0.15	
Age of diagnosis					0.80					>0.90					0.60
<60	58 (22%)	0.22	0.08	0.13		55 (24%)	0.22	0.07	0.12		55 (23%)	0.22	0.08	0.12	
≥60	203 (78%)	0.14	0.07	0.13		175 (76%)	0.14	0.06	0.13		181 (77%)	0.14	0.06	0.12	
Gleason					<0.01					<0.01					<0.01
<7	85 (33%)	0.21	0.08	0.15		81 (35%)	0.21	0.08	0.15		71 (30%)	0.21	0.07	0.15	
7	133 (51%)	0.13	0.05	0.12		119 (52%)	0.13	0.05	0.12		127 (54%)	0.13	0.05	0.12	
>7	43 (16%)	0.15	0.07	0.13		30 (13%)	0.16	0.08	0.13		38 (16%)	0.15	0.06	0.13	
PSA					0.06					0.15					0.11
<4	13 (5.0%)	0.13	0.04	0.12		12 (5.2%)	0.13	0.04	0.11		10 (4.2%)	0.13	0.03	0.11	
4–10	143 (55%)	0.17	0.06	0.12		136 (59%)	0.17	0.06	0.12		131 (56%)	0.17	0.05	0.12	
>10	105 (40%)	0.16	0.08	0.14		82 (36%)	0.15	0.08	0.13		95 (40%)	0.15	0.07	0.13	
Prostatectomy					0.03					0.03					0.04
No	57 (23%)	0.16	0.09	0.15		37 (17%)	0.16	0.08	0.15		48 (21%)	0.16	0.09	0.14	
Yes	195 (77%)	0.16	0.06	0.12		187 (83%)	0.16	0.05	0.12		182 (79%)	0.16	0.05	0.12	
Unknown	9	NA	NA	NA		6	NA	NA	NA		6	NA	NA	NA	
Radiotherapy					>0.90					0.60					0.80
No	114 (49%)	0.18	0.06	0.12		107 (51%)	0.18	0.05	0.12		103 (48%)	0.18	0.06	0.12	
Yes	121 (51%)	0.14	0.06	0.12		104 (49%)	0.14	0.06	0.12		112 (52%)	0.14	0.05	0.12	
Unknown	26	NA	NA	NA		19	NA	NA	NA		21	NA	NA	NA	
Chemotherapy					0.30					0.09					0.60
No	207 (93%)	0.16	0.05	0.12		191 (95%)	0.16	0.05	0.12		191 (93%)	0.16	0.05	0.12	
Yes	16 (7.2%)	0.15	0.09	0.12		10 (5.0%)	0.16	0.08	0.16		14 (6.8%)	0.14	0.07	0.12	
Unknown	38	NA	NA	NA		29	NA	NA	NA		31	NA	NA	NA	
Hormonotherapy					0.03					0.04					0.04
No	154 (64%)	0.16	0.05	0.12		151 (71%)	0.16	0.05	0.12		142 (65%)	0.16	0.05	0.12	
Yes	86 (36%)	0.15	0.08	0.13		62 (29%)	0.15	0.08	0.13		77 (35%)	0.15	0.07	0.13	
Unknown	21	NA	NA	NA		17	NA	NA	NA		17	NA	NA	NA	

^1^ Kruskal–Wallis rank-sum test.

**Table 3 cancers-16-02618-t003:** Survival of prostate cancer patients according to cobalt serum levels.

	Death Frequency			Univariable COX Regression			Multivariable COX Regression		
Variable	Overall, *n* = 261	Alive, *n* = 205	Dead, *n* = 56	HR	95% CI	*p*	HR	95% CI	*p*
Cobalt level									
QI (reference): 0.06–0.10 (0.09)	59 (23%)	51 (25%)	8 (14%)	—	—		—	—	
QII: 0.10–0.12 (0.11)	58 (22%)	51 (25%)	7 (13%)	0.89	0.32–2.45	0.80	0.93	0.33–2.58	0.90
QIII: 0.12–0.15 (0.14)	62 (24%)	51 (25%)	11 (20%)	1.31	0.53–3.26	0.60	1.15	0.46–2.88	0.80
QIV: 0.15–4.02 (0.27)	82 (31%)	52 (25%)	30 (54%)	3.02	1.38–6.59	<0.01	2.60	1.17–5.82	0.02

**Table 4 cancers-16-02618-t004:** Survival of prostate cancer patients according to cobalt serum levels among non-cancer-related deaths.

	Death Frequency			Univariable COX Regression			Multivariable COX Regression		
Variable	Overall, *n* = 230	Alive, *n* = 205	Dead, *n* = 25	HR	95% CI	*p*	HR	95% CI	*p*
Cobalt level									
QI (reference): 0.06–0.10 (0.09)	54 (23%)	51 (25%)	3 (12%)	—	—		—	—	
QII: 0.10–0.12 (0.11)	53 (23%)	51 (25%)	2 (8.0%)	0.68	0.11–4.08	0.70	0.64	0.11–3.86	0.60
QIII: 0.12–0.15 (0.14)	56 (24%)	51 (25%)	5 (20%)	1.63	0.39–6.84	0.50	1.55	0.37–6.52	0.50
QIV: 0.15–4.02 (0.28)	67 (29%)	52 (25%)	15 (60%)	4.39	1.27–15.20	0.02	3.67	1.03–13.00	0.04

**Table 5 cancers-16-02618-t005:** Survival of prostate cancer patients according to cobalt serum levels among prostate cancer-specific deaths.

	Death Frequency			Univariable COX Regression			Multivariable COX Regression		
Variable	Overall, *n* = 236	Alive, *n* = 205	Dead, *n* = 31	HR	95% CI	*p*	HR	95% CI	*p*
Cobalt level									
QI (reference): 0.06–0.10 (0.09)	56 (24%)	51 (25%)	5 (16%)	—	—		—	—	
QII: 0.10–0.12 (0.11)	56 (24%)	51 (25%)	5 (16%)	0.99	0.29–3.43	>0.90	1.13	0.32–3.92	0.90
QIII: 0.12–0.15 (0.14)	57 (24%)	51 (25%)	6 (19%)	1.17	0.36–3.82	0.80	0.87	0.26–2.93	0.80
QIV: 0.15–4.02 (0.28)	67 (28%)	52 (25%)	15 (48%)	2.64	0.96–7.26	0.06	2.17	0.76–6.18	0.15

## Data Availability

Data supporting the reported results are available from the corresponding author upon request from all interested researchers.

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
