# Peer review of "Cobalt Serum Level as a Biomarker of Cause-Specific Survival among Prostate Cancer Patients"

_cancers, 2024, doi:10.3390/cancers16152618_

Round 1

Reviewer 1 Report

Comments and Suggestions for Authors

The authors should be congratulated for the interesting topic discussed, even though major revisions are required. 

Please the grammar must be checked. 

1.     The table and graphics are clearly described.

2.     Materials and methods are robust.

3.     Please, clarify the aim of the study.

4.     Please, discuss how these findings could benefit the clinical practice by examining other kinds of serum biomarkers, too. For this purpose, reading these papers could help the authors satisfy the request: https://doi.org/10.1002/pros.24422.

5.     Please, list the limits and limitations of this work

6.     In line 48, did the authors intend “cobalt” instead of “cadmium”?

7.     The authors should provide references when discussing the WHO suggestion that cobalt is probably involved in carcinogenesis.

8.     Please, check the language in some parts of the text

Comments on the Quality of English Language

Minor editing.

Author Response

Comments: The authors should be congratulated for the interesting topic discussed, even though major revisions are required. Please the grammar must be checked. 

Response: Thank you, we appreciate this. The grammar has been checked.

Comments 1: The table and graphics are clearly described.

Response 1: Thank you for your comment.

Comments 2: Materials and methods are robust.

Response 2: Thank you for your comment.

Comments 3: Please, clarify the aim of the study.

Response 3: The aim of the study has been clarified.
The aim of this prospective study was to investigate relationship between serum cobalt levels and survival among prostate cancer patients taking into consideration prostate-cancer specific deaths and non-cancer cause of death.

Comments 4: Please, discuss how these findings could benefit the clinical practice by examining other kinds of serum biomarkers, too. For this purpose, reading these papers could help the authors satisfy the request: https://doi.org/10.1002/pros.24422.

Response 4: Excerpt describing the benefit the clinical practice by examining other kinds of serum biomarkers has been added to the manuscript.

Comments 5: Please, list the limits and limitations of this work

Response 5: Limits and limitations of this work has been added to the manuscript.
Limitations in this study were the relatively small study group and relatively low numbers of events (deaths), especially in subgroups of non-cancer related deats (n=25) and prostate-cancer specific deaths (n=31). Moreover, individuals included to study came from the same geographical region - Polish population, which may not be directly reference to all prostate cancer patients.

Comments 6: In line 48, did the authors intend “cobalt” instead of “cadmium”?

Response 6: This mistake has been corrected.

Comments 7: The authors should provide references when discussing the WHO suggestion that cobalt is probably involved in carcinogenesis.

Response 7: The reference has been added.

Comments 8: Please, check the language in some parts of the text

Response 8: The language has been corrected.

Reviewer 2 Report

Comments and Suggestions for Authors

I have read the manuscript titled “Cobalt serum level as a biomarker of cause specific survival among prostate cancer patients.” The authors aimed to explore the potential relationship between cobalt levels and the survival of prostate cancer patients. Despite the interesting research topic, the authors may wish to revise their manuscript considering the following comments.

1. In the abstract section, line 49, please clarify the meaning of “poor prostate cancer.” Does it refer to the prognosis or the patients’ conditions themselves?

2. The introduction section needs thorough revision to emphasize the background and rationale for conducting this study. In the second paragraph, lines 65-72, the information provided is overly general. Since the main objective of this manuscript is to identify the possible correlation between cobalt levels and prostate cancer (PCa) survival, the authors are encouraged to specifically deliver epidemiological information concerning PCa and the potential harms or characteristics of cobalt. For instance, in line 65, “The main causes of death among cancer patients are…” should primarily focus on PCa patients.

3. In subchapter 2.1, for clarity, please define the detailed inclusion and exclusion criteria. Are both clinically significant PCa (csPCa) and non-csPCa included? In addition, since the authors mentioned that this study is prospective, please include a flow chart detailing the patient selection process, encompassing the diagnostic modality.

4. English editing is required to enhance clarity and readability. For instance, in lines 135-136, the authors may consider adjusting the structure as follows: “For the whole study group (n=261), prostate cancer patients with high cobalt levels have worse survival compared to those with low cobalt serum levels in both univariable (HR=3.02; 95% CI: 1.38-6.59; p=0.006) and multivariable (HR=2.60; 95% CI: 1.17-5.82; p=0.02) COX regression models.”

5. In the discussion section, the authors spent a majority of the space introducing the basic information about cobalt and rarely discussed their results and findings. Considering the professional readership of this journal, the authors are encouraged to deeply interpret their results and reduce the content of popular science. Therefore, I recommend a thorough revision of the discussion section to increase the scientific depth and support their arguments.

Comments on the Quality of English Language

English editing is recommended. Please also pay attention to the flow of the manuscript. I recommend ensuring smooth transitions in the introduction and discussion sections.

Author Response

Comments 1: In the abstract section, line 49, please clarify the meaning of “poor prostate cancer.” Does it refer to the prognosis or the patients’ conditions themselves?

Response 1: This section has been changed in the text.

Comments 2: The introduction section needs thorough revision to emphasize the background and rationale for conducting this study. In the second paragraph, lines 65-72, the information provided is overly general. Since the main objective of this manuscript is to identify the possible correlation between cobalt levels and prostate cancer (PCa) survival, the authors are encouraged to specifically deliver epidemiological information concerning PCa and the potential harms or characteristics of cobalt. For instance, in line 65, “The main causes of death among cancer patients are…” should primarily focus on PCa patients.

Response 2: The introduction section has been updated on information regarding prostate cancer specifically.

Comments 3: In subchapter 2.1, for clarity, please define the detailed inclusion and exclusion criteria. Are both clinically significant PCa (csPCa) and non-csPCa included? In addition, since the authors mentioned that this study is prospective, please include a flow chart detailing the patient selection process, encompassing the diagnostic modality.

Response 3: This information has been added in the text. All consecutive prostate cancer patients (confirmed by histopathological examination) were included into the study. The inclusion of patients in the study was not dependent on whether they were clinically significant PCa (csPCa) or non-csPCa cases. Prostate cancer patients with missing data in: “Gleason”, “PSA”, and “cause of death” categories were excluded from analysis.
Since patients were not selected, in our opinion, for this type of patients enrollment, flow chart is not needed.

Comments 4: English editing is required to enhance clarity and readability. For instance, in lines 135-136, the authors may consider adjusting the structure as follows: “For the whole study group (n=261), prostate cancer patients with high cobalt levels have worse survival compared to those with low cobalt serum levels in both univariable (HR=3.02; 95% CI: 1.38-6.59; p=0.006) and multivariable (HR=2.60; 95% CI: 1.17-5.82; p=0.02) COX regression models.”

Response 4: English editing has been applied.

Comments 5: In the discussion section, the authors spent a majority of the space introducing the basic information about cobalt and rarely discussed their results and findings. Considering the professional readership of this journal, the authors are encouraged to deeply interpret their results and reduce the content of popular science. Therefore, I recommend a thorough revision of the discussion section to increase the scientific depth and support their arguments.

Response 5: The discussion section has been updated. Due to fact, that results presented in following manuscript are first that addressed the topic and at the same time reveal the relationship between serum cobalt levels and survival in prostate cancer patients, therefore we were unable to discuss the results thoroughly. In the discussion section, we have added information about the limitations of our study.

Comments: English editing is recommended. Please also pay attention to the flow of the manuscript. I recommend ensuring smooth transitions in the introduction and discussion sections.

Response: English editing and smooth transitions in the introduction and discussion sections have been applied.

Reviewer 3 Report

Comments and Suggestions for Authors

In this study, the authors investigate the relationship between cobalt serum levels and PCa survival. Here are some questions:

1.       What is the definition of high cobalt level? If high cobalt level means QIV, it would be better to descript it in the first place?

2.       What is the relationship between cobalt serum level and clinical signatures of PCa? Authors should use t-text or Chi-square test to compare different groups.

3.       The authors generate Kaplan-Meier curve to show the tendency of poor survival in patients with QIV Co levels. The Long-rank test should be used to check statistical significance.

4.       How to explain for Co level can be a risk factor in all death and non-cancer specific death, but not in cancer specific death?

5.       The results and figure legend is too simple and should be improved.

6.       Some minor flaws, such as in Line 166, the title of figure 1 should add “for all cause of death”; the decimal is not unified. Such as the p value in line 136, Table 7, etc; In Table 6, in the unknown group (such as Prostatectomy), mean should be “NA”.

Author Response

Comments 1: What is the definition of high cobalt level? If high cobalt level means QIV, it would be better to descript it in the first place?

Response 1:  High cobalt level is the cobalt assigned to the quarter with the highest level (QIV). This has been changed in the manuscript.

Comments 2: What is the relationship between cobalt serum level and clinical signatures of PCa? Authors should use t-text or Chi-square test to compare different groups.

Response 2:  The relationship between serum cobalt levels and investigated clinical factors was compared using nonparametric Kruskal-Wallis test instead of t test (according to the fact, that some data distributions were different than normal and due to fact, that for several factors we compared more than 2 groups of quantitative values). Appropriate description of used method and calculated p-values have been added into the text and tables.

Comments 3: The authors generate Kaplan-Meier curve to show the tendency of poor survival in patients with QIV Co levels. The Long-rank test should be used to check statistical significance.

Response 3:  The log rank test has been performed, the results were placed on the charts.

Comments 4: How to explain for Co level can be a risk factor in all death and non-cancer specific death, but not in cancer specific death?

Response 4:  In fact, the aforementioned discrepancy is surprising. It can be speculated, that cobalt may be associated with functioning of the cardiovascular system. There are papers indicating that high cobalt levels are related to the occurrence of cardiovascular dysfunctions (e.g. thrombosis), which can result in death. In our study group almost 30% of non-cancer specific deaths were cardiovascular related. However, it should be noted, that number of non-cancer specific and prostate cancer specific deaths are relatively low (n= 25 and n=31 respectively).

Comments 5: The results and figure legend is too simple and should be improved.

Response 5:  The results section and figures’ legends have been updated.

Comments 6: Some minor flaws, such as in Line 166, the title of figure 1 should add “for all cause of death”; the decimal is not unified. Such as the p value in line 136, Table 7, etc; In Table 6, in the unknown group (such as Prostatectomy), mean should be “NA”.

Response 6:  All substitutions have been applied.

Reviewer 4 Report

Comments and Suggestions for Authors

The manuscript presents a very good scientific challenge, but the context presentation needs improvement. Below, I have identified the key areas that need to be enhanced.

The language level is scientific and technical, which is expected and acceptable in a scholarly context. Some sentences could be shortened or restructured for better readability.

1.      The reference list consists of 56 sources, only 2 of which are from the last three years. I suggest significantly supplementing the list with articles from 2021 and newer. This will help the authors maintain a balance between foundational and recent research.

2.      In the introduction, it is necessary to more clearly specify how each mentioned factor specifically contributes to survival. For example, when presenting statistics on various survival factors, it would be helpful to indicate the specific impact they have and how they interact with each other.

3.      Transitions between different topics (e.g., from statistics to causes of death) could be smoother. Currently, they appear somewhat abrupt. It is suggested to use transitional sentences that link different ideas and topics.

4.      In some places, references to the literature are indicated, but their integration into the text could be clearer. For example, more information could be inserted about what the cited sources specifically confirm or refute.

5.      Review other research fields related to prostate cancer. Do not focus solely on your research problem. This section of the manuscript should also provide a broader overview. At least briefly introduce 5-7 articles. I suggest starting with these sources:

·         GibbonsM., et al. Prostate cancer lesion detection, volume quantification and high-grade cancer differentiation using cancer risk maps derived from multiparametric MRI with histopathology as the reference standard, Magnetic Resonance Imaging, Volume 99, 2023, pp. 48-57, https://doi.org/10.1016/j.mri.2023.01.006.

·         Milonas, D., et al. The Significance of Prostate Specific Antigen Persistence in Prostate Cancer Risk Groups on Long-Term Oncological Outcomes. Cancers 2021, 13, 2453. https://doi.org/10.3390/cancers13102453

6.      At the end of the introduction, where the research direction regarding the effect of cobalt on survival is indicated, it would be useful to clearly state why the effect of cobalt is important and how it differs from other already studied factors. It is suggested to present a clearer research hypothesis or question that will be addressed in the article.

7.      Technical descriptions (e.g., used reagents and devices) could be presented in an additional table or appendix, making the main text clearer and easier to read.

8.      It should be clearly stated how the study participants were selected and provide information about the selection criteria. It would be helpful to indicate why the specific period (2009-2015) was chosen for the study and what factors might have influenced the results.

9.      The statistical analysis section could be more clearly divided into separate parts: descriptive statistics, univariate analysis, and multivariate analysis. Breaking down and describing each part more broadly would help the reader better understand how different stages of the analysis were performed.

10.  Explaining some statistical terms would help in understanding the analysis methods. For example, a brief explanation of what the COX proportional hazards model and Kaplan-Meier curve are could be included.

11.  When discussing the results, it would be useful to mention the study's limitations that may affect the interpretation of the results. For instance, were there any methodological limitations or other factors that could impact the reliability of the results?

12.  More details about the results obtained during the study that support the conclusions should be provided.

Author Response

Comments 1: The reference list consists of 56 sources, only 2 of which are from the last three years. I suggest significantly supplementing the list with articles from 2021 and newer. This will help the authors maintain a balance between foundational and recent research.

Response 1:  The list of references has been updated. However, to the best of our knowledge, there is lack of papers available describing the relationship between serum cobalt levels and survival of prostate cancer patients. We believe that our work is the first, and therefore we have not been able to provide extensive, literature focusing strictly on survival.

Comments 2: In the introduction, it is necessary to more clearly specify how each mentioned factor specifically contributes to survival. For example, when presenting statistics on various survival factors, it would be helpful to indicate the specific impact they have and how they interact with each other.

Response 2:  The introduction section has been updated.

Comments 3: Transitions between different topics (e.g., from statistics to causes of death) could be smoother. Currently, they appear somewhat abrupt. It is suggested to use transitional sentences that link different ideas and topics.

Response 3:  Appropriate changes have been made. 

Comments 4: In some places, references to the literature are indicated, but their integration into the text could be clearer. For example, more information could be inserted about what the cited sources specifically confirm or refute.

Response 4:  It has been changed.

Comments 5: Review other research fields related to prostate cancer. Do not focus solely on your research problem. This section of the manuscript should also provide a broader overview. At least briefly introduce 5-7 articles. I suggest starting with these sources:

  • GibbonsM., et al. Prostate cancer lesion detection, volume quantification and high-grade cancer differentiation using cancer risk maps derived from multiparametric MRI with histopathology as the reference standard, Magnetic Resonance Imaging, Volume 99, 2023, pp. 48-57, https://doi.org/10.1016/j.mri.2023.01.006.
  • Milonas, D., et al. The Significance of Prostate Specific Antigen Persistence in Prostate Cancer Risk Groups on Long-Term Oncological Outcomes. Cancers2021, 13, 2453. https://doi.org/10.3390/cancers13102453

Response 5:  We have added few articles into the manuscript.

Comments 6: At the end of the introduction, where the research direction regarding the effect of cobalt on survival is indicated, it would be useful to clearly state why the effect of cobalt is important and how it differs from other already studied factors. It is suggested to present a clearer research hypothesis or question that will be addressed in the article.

Response 6:  Since the Co alloys have been classified by European Chemicals Agency as a Class 1B Carcinogen (presumed to have carcinogenic potential for humans), we found it interesting to check whether Co levels can significantly differentiate prostate cancer patients in terms of survival. We also decided to include in our considerations the causes of death of patients enrolled to the herein study for broader perspective.

Comments 7: Technical descriptions (e.g., used reagents and devices) could be presented in an additional table or appendix, making the main text clearer and easier to read.

Response 7:  Some of the reagents and devices used are necessary to describe the "Measurement methodology" section, but we have tuned in to your comment and removed the data on the manufacturers of these materials and included them in the appendix (Table A1).

Comments 8: It should be clearly stated how the study participants were selected and provide information about the selection criteria. It would be helpful to indicate why the specific period (2009-2015) was chosen for the study and what factors might have influenced the results.

Response 8: Consecutive patients with prostate cancer were included in the study (this information was added to the manuscript), they were recruited after diagnosis, based on histopathological examination. These were the only inclusion criteria in the database.
The period 2009-2015 was determined by samples availability.

Comments 9: The statistical analysis section could be more clearly divided into separate parts: descriptive statistics, univariate analysis, and multivariate analysis. Breaking down and describing each part more broadly would help the reader better understand how different stages of the analysis were performed.

Response 9: The statistical analysis section has been changed.

Comments 10: Explaining some statistical terms would help in understanding the analysis methods. For example, a brief explanation of what the COX proportional hazards model and Kaplan-Meier curve are could be included.

Response 10: A brief explanation have been added in the manuscript. COX proportional hazard models are often used, when association between event outcome (and time of event occurrence) and predictors need to be established. Kaplan-Meier curves are the graphical method which enables to show the relationship between participants’ survival rate during follow-up. 

Comments 11: When discussing the results, it would be useful to mention the study's limitations that may affect the interpretation of the results. For instance, were there any methodological limitations or other factors that could impact the reliability of the results?

Response 11: Limitations have been added into the manuscript.

Comments 12: More details about the results obtained during the study that support the conclusions should be provided.

Response 12: More details have been added in the conclusion section.

Round 2

Reviewer 1 Report

Comments and Suggestions for Authors

Authors answered all comments and suggestions

Author Response

Comments 1: Authors answered all comments and suggestions

Response 1: Thank you for your thorough review.

Reviewer 2 Report

Comments and Suggestions for Authors

I have reviewed the revised manuscript and found that the authors have extensively modified their original draft in response to my comments. However, I still have some comments the authors may wish to consider to further improve the manuscript:

1. In line 99, "The aim if this" should be revised to "the aim of this...". Please carefully review the entire manuscript to rectify similar typographic issues.

2. While the authors have included additional information regarding the diagnosis and treatment options for prostate cancer, some state-of-the-art information is still missing.

    (1) In line 238, the authors stated, "Alternative methods such as cryotherapy or HIFU require further research." There are additional novel treatment modalities worth mentioning, such as immunotherapy (immune checkpoint inhibitors, CAR-T cell therapy, etc.), photothermal therapy, and photodynamic therapy.

    (2) In lines 245-246, the authors mentioned, "One such method is prostate cancer mapping using multiparametric MRI (mpMRI)." The Pi-RADS scoring system should be mentioned as it is already utilized in clinical practice. Moreover, biparametric MRI has been found to have comparable effectiveness in screening for csPCa.

    (3) In lines 263-268, the authors introduced PSMA-PET-CT. This entire paragraph should be moved forward to line 246 for better structure and flow, as PSMA-PET-CT is also an imaging tool. Additionally, several contrast agents targeting PSMA as novel molecular imaging modalities should be discussed.

    (4) In the introduction section, the authors indicated the influence of cobalt on survival at the intracellular level, such as DNA damage and ROS production. Recent research has found that ROS is connected with mitochondrial alterations and cellular death, such as ferroptosis. These interplay between cobalt and intracelluar actions are very interesting and worth a more detailed discussion in the discussion section.

Author Response

Comments 1: In line 99, "The aim if this" should be revised to "the aim of this...". Please carefully review the entire manuscript to rectify similar typographic issues.

Response 1: Manuscript has been reviewed and changes have been made.

Comments 2: While the authors have included additional information regarding the diagnosis and treatment options for prostate cancer, some state-of-the-art information is still missing.

    (1) In line 238, the authors stated, "Alternative methods such as cryotherapy or HIFU require further research." There are additional novel treatment modalities worth mentioning, such as immunotherapy (immune checkpoint inhibitors, CAR-T cell therapy, etc.), photothermal therapy, and photodynamic therapy.

    (2) In lines 245-246, the authors mentioned, "One such method is prostate cancer mapping using multiparametric MRI (mpMRI)." The Pi-RADS scoring system should be mentioned as it is already utilized in clinical practice. Moreover, biparametric MRI has been found to have comparable effectiveness in screening for csPCa.

    (3) In lines 263-268, the authors introduced PSMA-PET-CT. This entire paragraph should be moved forward to line 246 for better structure and flow, as PSMA-PET-CT is also an imaging tool. Additionally, several contrast agents targeting PSMA as novel molecular imaging modalities should be discussed.

    (4) In the introduction section, the authors indicated the influence of cobalt on survival at the intracellular level, such as DNA damage and ROS production. Recent research has found that ROS is connected with mitochondrial alterations and cellular death, such as ferroptosis. These interplay between cobalt and intracelluar actions are very interesting and worth a more detailed discussion in the discussion section.

Response 2: We have added all the following information in the manuscript.

Reviewer 3 Report

Comments and Suggestions for Authors

The authors make a lot of changes and are definitely improve the manuscript. However, there are still some other changes can be made. 

1. In the abstract, the authors claimed “found a significant relationship between high serum cobalt levels and poor prostate cancer patient survival”. It is better to clarified here the survival is total survival. 

2. Please rewrite the results about Table 2 with more details. Firstly, descript which variables are statistically significantly related with serum cobalt level, and which are not. Secondly descript the meaning of the significant relation. For examples, as shown in the table, the serum level is found correlated with Gleason scores. The correlation is positive or negative?

3. There are redundance of table 2,4,6. The authors can add an additional part in variable column and separate patients as total, cancer-specific, non-cancer-specific death, and compared the difference of the groups. In this way, these three tables can be combined as one.

4. In the variables, the authors found serum levels is correlated with patients’ therapy such as prostatectomy and chemotherapy. What is the reason for that? Since patients receive different therapy will absolutely influence their survival, does serum level have relationship with survival in patients with same therapy?

Author Response

Comments 1: In the abstract, the authors claimed, “found a significant relationship between high serum cobalt levels and poor prostate cancer patient survival”. It is better to clarified here the survival is total survival.

Response 1: Correction has been made.

Comments 2: Please rewrite the results about Table 2 with more details. Firstly, descript which variables are statistically significantly related with serum cobalt level, and which are not. Secondly descript the meaning of the significant relation. For examples, as shown in the table, the serum level is found correlated with Gleason scores. The correlation is positive or negative?

Response 2: The results section related to Table 2 have been updated. We have been provided in text description of variables for which serum cobalt levels were significantly different. Since the Gleason score was categorized as <7; 7 and >7, we calculated Spearman’s rank correlation coefficient. According to the results it can be stated that there is a weak, negative correlation (rs=-0.13; p=0.04) between serum cobalt levels and Gleason score. Due to fact that Gleason variable is analyzed in 3 categories, we decided not to include information on Spearman correlation analysis in the text, leaving information on median serum cobalt levels for each Gleason category in Table 2 (medians were compared using non-parametric Kruskal-Wallis test – information included in the main text).

Comments 3: There are redundance of table 2,4,6. The authors can add an additional part in variable column and separate patients as total, cancer-specific, non-cancer-specific death, and compared the difference of the groups. In this way, these three tables can be combined as one.

Response 3: Tables 2, 4 and 6 has been merged into one (Table 2).

Comments 4: In the variables, the authors found serum levels is correlated with patients’ therapy such as prostatectomy and chemotherapy. What is the reason for that? Since patients receive different therapy will absolutely influence their survival, does serum level have relationship with survival in patients with same therapy?

Response 4: All blood samples have been collected shortly after diagnosis and before treatment. However, we decided to make the calculations regarding the relationship between serum cobalt levels and overall survival among those study participants who received the same therapy e.g. on subgroup consisted only of men after prostatectomy or those who received only hormonotherapy (not chemotherapy - serum cobalt levels were not statistically different among patients who received and not received chemotherapy p=0.30; p=0.09 and p=0.60 for all study group, non-cancer related deaths and prostate cancer specific deaths respectively). We found that prostate cancer patients with high serum cobalt levels, who were undergoing a prostatectomy have significantly (p = 0.04) worse survival, however, due to relatively small number of all cause death events (n = 12) we did not decide to put those results into main text. For the same reason, analyses on non-cancer related deaths and prostate specific deaths only were not performed. According to study participants who receive only hormonotherapy, there were only 44 such patients in total. In connection to the above we did not perform regression analysis on this subgroup at all.

Reviewer 4 Report

Comments and Suggestions for Authors

The manuscript is significantly improved. The authors took into account the recommendations presented. I believe that the manuscript can be accepted for publication.

Author Response

Comments 1: The manuscript is significantly improved. The authors took into account the recommendations presented. I believe that the manuscript can be accepted for publication.

Response 1: Thank you for your thorough review.

Round 3

Reviewer 3 Report

Comments and Suggestions for Authors

It is very happy to see authors made all edits, which improve a lot of the manuscript. 

Please add all the missing parts (including Author contribution, funding, references, etc.) after conclusion. Thanks!